# Circular Optical Phased Array with Large Steering Range and High Resolution

**DOI:** 10.3390/s22166135

**Published:** 2022-08-16

**Authors:** Daniel Benedikovič, Qiankun Liu, Alejandro Sánchez-Postigo, Ahmad Atieh, Tom Smy, Pavel Cheben, Winnie N. Ye

**Affiliations:** 1Department Electronics, Carleton University, Ottawa, ON K1S 5B6, Canada; 2Department Multimedia and Information-Communication Technology, University of Zilina, 01026 Zilina, Slovakia; 3University Science Park, University of Zilina, 01026 Zilina, Slovakia; 4Telecommunication Research Institute (TELMA), Universidad de Málaga, E.T.S. Ingeniería de Telecomunicación, Bulevar Louis Pasteur 35, 29010 Málaga, Spain; 5Optiwave Systems, Inc., Ottawa, ON K2E 8A7, Canada; 6National Research Council Canada, Ottawa, ON K1A 0R6, Canada

**Keywords:** silicon photonics, optical phased arrays, optical antenna, beam forming, beam steering, field-of-view, circular phased arrays, light detection and ranging

## Abstract

Light detection and ranging systems based on optical phased arrays and integrated silicon photonics have sparked a surge of applications over the recent years. This includes applications in sensing, free-space communications, or autonomous vehicles, to name a few. Herein, we report a design of two-dimensional optical phased arrays, which are arranged in a grid of concentric rings. We numerically investigate two designs composed of 110 and 820 elements, respectively. Both single-wavelength (1550 nm) and broadband multi-wavelength (1535 nm to 1565 nm) operations are studied. The proposed phased arrays enable free-space beam steering, offering improved performance with narrow beam divergences of only 0.5° and 0.22° for the 110-element and 820-element arrays, respectively, with a main-to-sidelobe suppression ratio higher than 10 dB. The circular array topology also allows large element spacing far beyond the sub-wavelength-scaled limits that are present in one-dimensional linear or two-dimensional rectangular arrays. Under a single-wavelength operation, a solid-angle steering between 0.21*π* sr and 0.51*π* sr is obtained for 110- and 820-element arrays, respectively, while the beam steering spans the range of 0.24*π* sr and 0.57*π* sr for a multi-wavelength operation. This work opens new opportunities for future optical phased arrays in on-chip photonic applications, in which fast, high-resolution, and broadband beam steering is necessary.

## 1. Introduction

Light detection and ranging (LIDAR) has become a core technology for many applications, including environmental monitoring and sensing [1,2], imaging [3], free-space communications [4], and autonomous vehicles [5]. Despite their diverse deployment, traditional LIDAR systems rely on a large amount of discrete optical elements and equipment and, in addition, depend on mechanical mechanisms to fulfil desired functions of beam steering and shaping [6]. These factors promote high production cost, large size, excessive power consumption, and increased complexity while providing only limited steering range, low resolution, and slow scanning rates. From a long-term perspective, with ever-increasing needs for reliability and scalability, bulky and mechanically rotating approaches are difficult to integrate and scale and will inevitably fail to meet the requirements of driving applications. 

In contrast, foundry-based integrated silicon photonics offer great potential for chip-scale LIDARs [7,8,9]. Integrated photonics leverages the maturity of semiconductor microelectronics and the recent advances in high performance photonic devices to achieve true monolithic photonic–electronic integration on a single chip. 

In the last decade, LIDAR systems based on integrated optical phased arrays (OPAs) have taken a massive leap forward [10,11,12,13,14,15,16,17,18,19,20,21,22,23,24,25,26,27,28,29,30]. On-chip OPAs can empower solid-state beam steering, allowing for high-resolution and fast scanning. For the most part, OPAs implemented on silicon photonic platforms, silicon-on-insulator (SOI) in particular, have been designed as one-dimensional (1-D) linear [10,11,12,13,14,15,16] or two-dimensional (2-D) rectangular [4,17,18,19,20,21,22,23,24,25,26] arrays. In these array configurations, beam steering can be achieved through wavelength-tuned millimeter-long antennas [4,25] or by using optical antennas with cascaded- or independently-controlled phase shifter arrangements [21,22]. Although useful techniques for beam steering have been demonstrated using 1-D linear and 2-D rectangular topologies, a range of performance and design issues remain. For example, the standard OPA far-field radiation pattern exhibits large sidelobes as a consequence of the uniform array distribution. The large sidelobes appear when the spacing between two adjacent optical elements is larger than a half of the operating wavelength. This is, however, a common concern in many photonic technologies, regardless of which of the afore-mentioned OPA arrangements is employed. As a result, 1-D linear and 2-D rectangular arrays typically impose limitations on an aliasing-free beam steering and suffer from a restricted field of view (FOV). 

To remove the grating sidelobes and improve the beam-steering performance, circular OPAs have been proposed and represent an important choice [31,32,33,34,35,36,37,38,39,40,41]. Surprisingly, only limited preliminary work on photonic implementation of circular array has been reported to date [39,40,41], apart from bio-inspired solutions [31], non-photonic-based concepts [32], and optimization-driven circular schemes [33,34,35,36,37]. Indeed, compared to the conventional 1-D/2-D counterparts, the use of circular arrays holds great promise to advance the overall steering performance by improving the FOV ideally up to 2*π* sr. 

In this work, we report a comprehensive study on a 2-D OPA using a concentric-ring configuration. More specifically, we numerically investigate two circular array designs, one with 110 elements (*Array #1*) and another with 820 elements (*Array #2*). The proposed circular OPAs are designed to provide beam-steering functions across the *C*-band communication window, ranging from 1535 nm to 1565 nm. Both circular OPAs show excellent beam-steering performance, maintaining a sidelobe suppression level larger than 10 dB, with narrow angular beamwidths of 0.5° (*Array #1*) and 0.22° (*Array #2*). Moreover, circular topology allows the OPA layouts to be implemented with a large element spacing of 9 μm. Unlike in previous developments, our circular arrays utilize surface-emitting antennas with high radiation efficiency and broadband operation, which allows us to operate in a multi-wavelength regime to further increase the OPA steering range. Solid-angle steering of 0.21*π* sr (*Array #1*) and 0.51*π* sr (*Array #2*) is obtained for monochromatic operation at a reference wavelength of 1550 nm, while improved steering ranges of 0.24*π* sr (Array #1) and 0.57*π* sr (Array#2) are obtained for a multi-wavelength case. This work opens the door for the development of the next-generation OPA systems on a single silicon chip that offer wideband, high-resolution, and fast beam steering.

## 2. Circular Array Geometry and Array Factor

A schematic diagram of the proposed circular array is shown in Figure 1a. The green spheres represent individual nano-antennas. The antenna layout is shown in Figure 1b. The circular array consists of *N* concentric rings, and the radius of the respective ring is given by:(1)$$R_{n}=R_{0}+\left(n-1\right)dr$$
where *R*_0_ is the minimum radius of the initial ring, *n* is the ring number, and *dr* denotes the radius increment between two adjacent rings. On each ring, there are *M* antennas with equal angular spacing between two adjacent elements. In total, *M* × *N* elements are radially arranged in the circular array. As we detailed in Ref. [41], the feeding stage of the circular OPA system comprises an efficient input optical interface formed by a circular input grating coupler [42], followed by short adiabatic waveguide tapers, and output waveguides that connect the phase shifters with individual optical antennas. In particular, the central input coupler consists of a blazed waveguide grating with an *L*-shaped radiation profile and sub-wavelength grating (SWG) metamaterial blocks. The grating coupler design is optimized to match the upward radiation beam to the near-Gaussian profile of the standard single-mode optical fiber (SMF-28). The optical fiber has a mode field diameter of 10.4 μm at 1/e^2^ intensity and nominal wavelength of 1550 nm. The coupling efficiency of the central grating coupler was calculated as a product between the power radiated from the grating and an overlap between the near-field profile of upward radiated beam and a profile of the fiber mode. According to our calculations, the *L*-shaped central grating coupler radiates 86.3% of input power towards an optical fiber situated above the chip, while only 3.8% is diffracted into the bottom silicon substrate. The near-field overlap between the radiated grating beam and the optical fiber mode is estimated to be 76%. As a result, the coupling loss of the central input waveguide coupler is −1.83 dB, calculated for a quasi-transverse electrical polarization and an operating wavelength of 1550 nm.

In the following numerical study, we assume isotropic nano-antennas where the signal power is emitted uniformly in all directions. For a circular OPA, the array factor is defined as follows [43,44]: (2)$$A\left(\theta ,\phi \right)=\sum_{m=1}^{M}\sum_{n=1}^{N}a_{mn}e^{-i\frac{2\pi }{\lambda }\sin\left(\theta \right)\cos\left(\phi -\frac{2\pi \left(m-1\right)}{M}\right)R_{n}}$$
where the element weight coefficient is *a**_mn_* = 1/(*M* × *N*), *λ* is the wavelength, and *θ* and *Φ* are the elevation and azimuthal angles, respectively. 

To systematically study the performance of the circular array, we calculated the angular beamwidth and sidelobe suppression as a function of four key parameters that determine the array factor: *N*, *M*, *R*_0_, and *dr*. In this work, the angular beamwidth and sidelobe suppression are defined as follows: (i.) the angular beamwidth corresponds to the full width at half maximum (FWHM) of the main lobe and (ii.) the sidelobe suppression expresses the difference between the peak of the main lobe, which is normalized to 0 dB, and the maximum of all other peaks in the scanning range. For numerical simulations, a resolution of 0.01° is considered. The array factor is calculated for a reference wavelength of 1550 nm. 

Figure 2 shows the results of a comprehensive study on the effect of the key array parameters (*N*, *M*, *R*_0_, and *dr*) on the circular OPA beamwidth and sidelobes. To determine the array geometry, we varied a selected parameter, while the others remained constant, and assessed the trends in the array performance metrics:*Angular beamwidth.* In Figure 2a,c,d, the angular beamwidth of the array factor’s main lobe decreases exponentially as the number of rings (*N*) rises, the minimum ring radius (*R*_0_) increases, and the spacing between two adjacent rings (*dr*) expands. In contrast, the number of the optical antennas per single ring (*M*) has a minimum effect (Figure 2b), and thus the angular beamwidth remains virtually constant, or, more likely, its variation is too small to be distinguished with a 0.01° resolution.*Sidelobe suppression ratio (SSR).* The sidelobe suppression exponentially increases with the number of concentric rings (*N*) within an array (see Figure 2a). Moreover, as shown in Figure 2b, the sidelobe suppression increases with the number of elements per ring (*M*). On top of that, the dependence for even and odd element numbers is apparent. When *M* is even, as shown by stars in Figure 2b, the circular array is bilaterally symmetric at any direction in the *x-y* plane, and thus constructive interference of all elements dominates. The sidelobes of each sub-array rise at similar {*θ*,*Φ*} angles, leading to a slightly reduced main-to-sidelobe contrast level. This effect is shown in Figure 3a,b, with the number of elements per single ring being *M* = 4 and *M* = 8, respectively. Here, the level of the sidelobe suppression remains small. The bilateral symmetry is broken when *M* is odd (shown by dots in Figure 2b). The broken symmetry helps to reduce the magnitude of the grating sidelobes. As a result, the power is distributed more uniformly over the considered angular range with a distinct sidelobe suppression. Last, but not least, the sidelobe suppression is not strongly affected by variations in the minimum ring radius (*R*_0_) (see Figure 2c) as well as in the ring spacing (*dr*) (see Figure 2d). For example, with *R*_0_ ranging from 0 to 100 μm and *dr* ranging from 0 to 10 μm, the sidelobe suppression varies from a minimum of about 8 dB up to a maximum of 12 dB. These low variations are likely attributed to the fact that changes in *R*_0_ and *dr* do not substantially affect the symmetry of the array and have a minimal effect on the element periodicity.

Motivated by the calculations above, we set the following design considerations: a target sidelobe suppression level of −10 dB, a minimum angular beamwidth of 1°, and an inter-element separation (*dr*) much greater than a half of an operating wavelength to make room for phase shifters [7,8,9,10]. According to our analysis, we propose two different circular phased arrays denoted as *Array #1* and *Array #2*, consisting of 110 and 820 elements, respectively. The design parameters of both circular arrays are provided in Table 1.

Figure 3c shows an elevation cut *θ* of an array factor with *Φ* = 0 for *Array #1*. We can observe a sidelobe suppression of −11.1 dB, and the estimated angular beamwidth of this circular array is 0.5°. The enlarged view of the main lobe is shown in the inset of Figure 3c. The sidelobe suppression level and the angular beamwidth can be further improved by increasing the total number (*M* × *N*) of OPA elements. Figure 3d shows an elevation cut *θ* of an array factor for *Array #2* with 820 elements. The obtained main-to-sidelobe contrast is substantially improved to −19.2 dB. Furthermore, the increased *N* of this array also provides a narrower angular beamwidth of 0.22°, as shown in the inset of Figure 3d. For autonomous vehicle applications, these array features can recognize a 38 cm large object at a distance of about 100 m from the OPA. 

Figure 4a shows the array factor for *Array #1* in the *uv* space. The parameters *u* and *v* relate to the elevation and azimuthal angles as follows: *u* = sin (*θ*)cos (*Φ*) and *v* = sin (*θ*)sin (*Φ*). For the numerical analysis, we considered 1° resolution over the scanning region, corresponding to an entire hemisphere (elevation angle: *θ* = 0°:1°:90°, and azimuthal angle: *Φ* = 0°:1°:360°). 

To demonstrate the advantages of the circular array topology, in Figure 4 we compare the array factor of a circular array (*Array #1*) with the array factor of a rectangular configuration, both having an identical number of elements and separation distance. For an array with a rectangular layout, the array factor is defined as [43,44]:(3)$$A\left(\theta ,\phi \right)=\sum_{m=1}^{M}\sum_{n=1}^{N}a_{mn}e^{-i\frac{2\pi }{\lambda }\left[d_{x}\sin\left(\theta \right)\;\cos\left(\phi \right)+d_{y}\sin\left(\theta \right)\;\sin\left(\phi \right)\right]}$$
where *M* and *N* denote the number of elements in each row and each column, respectively, and *d_x_* and *d_y_* are element spacings along *x* and *y* axis, respectively. 

We note that in this rectangular array, the element spacing is set to be the minimum distance between adjacent elements in the circular array, i.e., *d_x_* = *d_y_* = *dr*. As shown in Figure 4a, only the main lobe is visible for a circular array configuration, even though the element spacing is substantially larger than a half of an operating wavelength. On the contrary, large sidelobes are periodically distributed and are consistently dominant over the full scanning region as illustrated in the corresponding rectangular array configuration of Figure 4b. The periodic arrangement of the array and a rectangular element spacing in both directions (which is indeed larger than a half of a wavelength) are responsible for the presence of strong sidelobes and for the low sidelobe suppression ratio.

## 3. Far-Field Pattern of Circular Array

The array far-field pattern (AFP) of the circular array is defined as follows:(4)$$\mathrm{AFP}=E\left(\theta ,\phi \right)\times A\left(\theta ,\phi \right)$$
where *E* (*θ*,*Φ*) is the far field of an individual antenna and *A* (*θ*,*Φ*) is the array factor. For practical OPAs, the actual steering range is limited by the grating lobes of the array factor *A* (*θ*,*Φ*) and by the far-field pattern of the radiation element, i.e., surface-emitting antenna *E* (*θ*,*Φ*). For typical rectangular arrays, the presence of grating lobes is the main restriction as distance between individual elements is much larger than a half of a wavelength. Therefore, the actual steering range of rectangular arrays can be represented by the elevation and azimuthal angles obtained by the angular location of the grating lobes. For instance, for the rectangular configuration, as shown in Figure 4b, the beam steering is limited to only 9.9° × 90° (*θ* × *Φ*). In contrast, for arrays with a circular configuration, the grating lobes are effectively suppressed with respect to the main lobe. The beam-steering range is then determined by the far-field pattern of the grating antenna.

To quantify the steering range, we calculate the solid angle Ω*_SR_* as:(5)$$\Omega_{SR}=\iint_{SR}\sin\left(\theta \right)d\theta d\phi$$

The Ω*_SR_* ranges from 0 to 2*π*, corresponding to the solid angle of a full hemisphere. 

The circular array can be treated as *N* sub-arrays with *M* elements, having an element spacing of *R*_n_sin (2*π*/*M*). The grating sidelobes of each sub-array arise at different elevation and azimuthal angles {*θ*, *Φ*}. The overall array factor is represented by the convolution of an array factor of respective *N* sub-arrays. Due to this, the constructive interference remains dominant only at the main lobe at the elevation and azimuthal angles {*θ*_A_, *Φ*_A_}. As a result, the circular array can provide a maximum theoretical FOV of 90° × 360° (*θ* × *Φ*), even with an element spacing that is larger than a half of a wavelength. This steering range leads to a solid angle of 2*π* sr of a full hemisphere. It is worth noting that the actual FOV remains restricted by the specific radiation pattern of the individual antennas.

## 4. Antenna Radiation Pattern

In an ideal case, the circular array is formed by isotropic antennas, which ideally yields a maximum theoretical FOV of 2*π* sr. However, designing such an antenna is not practical. Our proposed optical antenna, schematically shown in Figure 1b, was designed on an SOI platform with 220 nm thick Si waveguide and 2 μm thick buried oxide (BOX) layers. For the upper cladding, we used a silicon dioxide (SiO_2_) layer. 

The Si antenna has an *L*-shaped grating profile [45,46,47,48,49] that leverages two asymmetric scatterers, resulting in constructive and destructive interference in the upward and downward direction, respectively. The two-level blazed profile of the optical antenna maximizes radiation efficiency and directionality while at the same time facilitating a broadband device operation. For clarity, the antenna radiation efficiency is defined as the optical power radiated upwards normalized to the input power launched into the waveguide, and the antenna directionality is defined as the ratio between the power radiated upwards and the total out-coupled power (power diffracted upwards and downwards). Using finite difference time domain (FDTD) calculations [50], the diffraction grating was designed to operate with an in-plane input light polarization, i.e., quasi-transverse electric (quasi-TE) polarization. As illustrated in the inset of Figure 1b, along the light propagation direction, the grating period is formed with four radiating segments. The segments are denoted as follows: deep-etch SWG trench (*L*_swg_), deep-etch oxide trench (*L*_d_), shallow-etch Si trench (*L*_s_), and unetched Si tooth (*L*_n_). Antenna profile with *L*-shaped geometry is defined through full- and partial etches with depths of 220 nm and 110 nm, respectively, as both steps are available in standard open-access foundries and multi-project wafer offerings [51,52]. The surface-emitting antenna has five periods, each *Λ*_a_ = *L*_swg_ + *L*_d_ + *L*_s_ + *L*_n_ long. The first grating trenches (*L*_swg_) are formed by uniform sub-wavelength grating (SWG) nano-structure [53,54,55] to reduce the return-loss at the waveguide-to-grating interface. The employed SWG geometry synthesizes an equivalent refractive index of 2.32 [41,45,46]. As shown in Figure 1b, the surface-emitting antenna has a curved layout with a total footprint of 2.5 μm × 5.5 μm (width × length). 

Figure 5a shows the distribution of the simulated electric field, in particular the real part of *E*_y_ field component. Indeed, due to the use of a blazed grating geometry, only a small amount of input power is radiated towards the Si substrate (less than 6% at 1550 nm). Compared to optical antennas implemented on 300 nm thick SOIs [48,56], surface gratings made in a standard 220 nm Si platform have reduced scattering strength per cell unit, and thus the residual transmitted power at the waveguide end remains higher, in this design case, at about 18%. The antenna far-field pattern, with light emission at 1550 nm, is shown in Figure 5b. The emission angle at 1550 nm is {*θ*_0_ = 6°, *Φ*_0_ = 0°}. Figure 5c shows a simulated out-radiated optical power from the surface-emitting antenna as a function of the wavelength. At 1550 nm, the radiation efficiency and the directionality of an *L*-shaped optical antenna are −1.49 dB (~71%) and −0.36 dB (~92%), respectively. By varying the input wavelength, the optical beam radiated towards an upward direction can be rotated in a 20° range, specifically from −9° (*θ*_0_ = 9°, *Φ*_0_ = 180°) at 1400 nm to 11° (*θ*_0_ = 11°, *Φ*_0_ = 0°) at 1600 nm.

## 5. Beam Formation and Beam-Steering Performance

The circular OPAs are formed such that all the nano-antennas have the same orientation along the *x*-axis. By knowing the OPA layout and antenna far-field pattern we can investigate the beam-steering performance of the OPA. In the following, we first discuss the beam steering of circular OPAs for a monochromatic operation, i.e., the single-wavelength operation, followed by broadband multi-wavelength operation. 

In this comprehensive study, we performed numerical simulations of the AFP at different steering angles {*θ*_A_, *Φ*_A_} over the entire hemisphere. The resolution used in our simulations was set to 1° (*θ* = 0°:1°:90°, *Φ* = 0°:1°:360°). Figure 6 shows the top view of the calculated AFP in a *uv* space for *Array #1*, operating at a 1550 nm wavelength with steering *θ*_A_ = 0° and *Φ*_A_ = 0°, resulting in a radiated beam directed to {*θ* = *θ*_0_ = 6°, *Φ* = *Φ*_0_ = 0°}.

### 5.1. Monochromatic Array Operation

In the single-wavelength operation, the antenna radiation pattern, *E*(*θ*, *Φ*), remains static, and the steering range of the whole OPA system is limited to the range defined by this pattern. By tailoring the phase of each radiation element via phase shifters, the steering angles {*θ*_A_, *Φ*_A_} can vary over the entire hemisphere. The steering range is determined as the set of steerable angles {*θ*, *Φ*}, for which the level of sidelobe suppression is larger than 10 dB. For the calculated far-field pattern of *Array #1* (shown in Figure 6) and *Array #2*, the sidelobe suppression levels are 12.3 dB and 18.4 dB, respectively. To determine the steering range, all steerable angles are projected on a *uv* plane. Figure 7 shows the projection of the steering range onto a hemisphere for an *Array #1* and an *Array #2*. Both arrays operate at a reference wavelength of 1550 nm. The yellow area represents the set of all steerable angles, with a sidelobe suppression larger than 10 dB. The blue areas, on the other hand, correspond to non-steerable angles. By using Equations (4) and (5), we estimated Ω*_SR_* of 0.21*π* sr for *Array #1* and 0.51*π* sr for *Array #2*, respectively.

### 5.2. Multi-Wavelength Array Operation

We now study the steering range in the multi-wavelength operation. As indicated in the previous sections, OPAs with a circular configuration can provide a maximum FOV of 2*π* sr, and according to Equation (5), the beam-steering range is only limited by the antenna radiation pattern. By wavelength tuning, the antenna radiation pattern can be rotated in the elevation direction. Consequently, the steering range is determined not only by the changes in {*θ*_A_, *Φ*_A_} via phase adjustments, but also by changes in {*θ*_0_, *Φ*_0_} via wavelength tuning. A maximum sidelobe contrast can be obtained when the peak angles of the array factor and the antenna far-field pattern coincide, i.e., *θ*_A_ = *θ*_0_ and *Φ*_A_ = *Φ*_0_. Unlike in the monochromatic operation, the beam steering is now optimized in two steps: (i.) by tuning the wavelength such that peak of antenna far-field pattern (*θ* = *θ*_0_, *Φ* = 0° for *λ* > 1490 nm; *θ* = *θ*_0_, *Φ* = 180° for *λ* < 1490 nm) is as close as possible to the target elevation for achieving the largest main-to-sidelobe contrast and (ii.) by modifying the phase of each individual antenna to steer the array factor’s main lobe to the target elevation and azimuthal angles. 

The study was performed for the *C*-band wavelengths ranging from 1530 nm to 1565 nm. The steerable angles are obtained by determining the level of the sidelobe suppression for the array far-field pattern, which is calculated at different steering angles and wavelengths. Figure 8a,b show the projection of the steering range into a *uv* space for *Array #1* and *Array #2*, yielding solid angles Ω*_SR_* of 0.24*π* sr and 0.57*π* sr, respectively, over the *C*-band spectral range. 

## 6. Power Consumption and Optical Loss in Circular Array

Besides the simplistic OPA layout and geometry and their promising beam-steering function, high-performance phased arrays call upon low-power-consuming elements for phase adjustments and low-loss passive building blocks for light coupling, guiding, and routing on a chip. Thus, for large-scale OPAs realized on a silicon platform, the power consumption and losses are also important design considerations. In the following, we estimate the power consumption of the proposed circular OPAs and we sum up the optical losses for the key passive elements of the proposed circular OPAs.

In waveguide-integrated OPAs, the 2-D beam steering is typically realized by using a phase tuning of waveguides and/or wavelength tuning of an input light source hand-in-hand with individual optical nano-antennas [10,11,12,13,14,15,16,17,18,19,20,21,22,23,24,25,26]. In our OPAs, where the optical nano-antennas are arranged in a grid of concentric rings, thermo-optic phase shifters were considered. They consist of a resistive material situated on top of/near a silicon strip waveguide. Thermal-based phase shifters are easy-to-implement devices, fully compatible with the open-access silicon foundry processing schemes and standard back-end-of-line (BEOL) metallization steps [51]. The efficiency of a thermal-based phase shifter can be calculated as follows [57,58,59]:(6)$$P_{\pi }=\Delta T_{\pi }\cdot G\cdot A$$
where *G* is thermal conductance, *A* is a waveguide cross-section, and Δ*T*_π_ refers to the temperature change to induce a *π* phase shift. In our case, the strip silicon waveguide is implemented as the phase shifter, having a rectangular cross-section of 220 nm by 500 nm (thickness by width). The temperature shift is defined as [58]:(7)$$\Delta T_{\pi }=\frac{\lambda }{2\cdot L\cdot dn/dT}$$

Here, *λ* is the operating wavelength (1550 nm, in our design case), *L* is the length of the silicon strip waveguide for the thermo-optic phase shifter, and *dn/dT* denotes the thermo-optic coefficient of silicon (*dn/dT* = 1.804e − 4 K^−1^ at a wavelength of 1550 nm and the nominal ambient temperature of 300 K [60]).

Adopting the analysis and data reported in Refs. [58,59], we estimated the power consumption of a single optical phase shifter as well as the overall power consumption of the proposed OPA design configurations. To this end, we considered different temperature shifts in the waveguide core to induce a *π* shift by the thermo-optic phase shifter implemented with silicon strip waveguide of different lengths. The temperature shift *ΔT* was varied in a 5 K to 20 K range. The lengths of the silicon strip waveguide phase shifter were varied between 1 μm and 10 μm, chosen to be consistent with our analysis on the OPA described in Section 2.

Figure 9 shows the estimated power consumption as a function of the waveguide length for different temperature shifts *ΔT*. From the calculated results, it becomes apparent that the power consumption decreases exponentially with an increasing waveguide length. In particular, Figure 9a shows a power consumption assessed for a single thermo-optic phase shifter. As a nominal showcase, for a 5 μm long phase shifter and temperature change equal to 10 K, the estimated power consumption is 17 mW. Accordingly, as shown in Figure 9b,c, the overall power consumption of the proposed circular OPAs comprising 110 and 820 elements is estimated to be 1.9 W and 14 W, respectively. It is evident that shorter (longer) thermo-optic phase shifters can enlarge (reduce) the overall power consumption of the OPA system. For instance, a 1 μm long phase shifter consumes 430 mW of power. This is indeed a prohibitively large amount of dissipated power in the system for a single phase-controlled phased array element [4,7,15,16]. Correspondingly, the overall consumption of the circular OPAs will reach excessive power levels of 47 W (*Array #1*) and 350 W (*Array #2*), respectively. In contrast, a thermo-optic phase shifter with a length of 10 μm helps to reduce the power consumption down to 4 mW. Consequently, the overall consumption of 110 and 820 circular OPAs can be lowered to power levels of 470 mW and 3.5 W, respectively.

The critical passive elements for light coupling, guiding, and routing of the circular OPA system include [41,61] the central input grating coupler, the optical nano-antennas, the adiabatic waveguide tapers, and the interconnecting waveguides. The summary of the losses for the key passive OPA elements is provided in Table 2.

## 7. Conclusions

We proposed and provided a comprehensive study of 2-D OPAs in a circular configuration. Specifically, we numerically investigated two circular array configurations, with 110 elements (*Array #1*) and 820 elements (*Array #2*), for both monochromatic (at 1550 nm) and multi-wavelength operations (in a 1535 nm to 1565 nm range). Circularly arranged OPAs showed substantial improvement over the traditional 1-D linear and 2-D rectangular arrays in beam-steering performance, with a sidelobe suppression contrast larger than 10 dB and with narrow angular beamwidths of 0.5° (for 110-element array) and 0.22° (820-element array). Considering an identical number of antenna elements, circular OPAs enabled an enlarged element spacing of 9 μm compared to their rectangular OPA counterparts. For the nominal 1.55 μm wavelength, the circular OPAs yield solid-angle steering ranges of 0.21*π* sr (for *Array #1*) and 0.51*π* sr (for *Array #2*). When the broadband operation was considered, this was further enhanced to 0.24*π* sr and 0.57*π* sr, respectively. These results are promising for the next-generation OPA systems realized on a silicon photonic platform with attractive performances in wideband, high-resolution, and fast free-space beam steering.

## Figures and Tables

**Figure 1 sensors-22-06135-f001:**
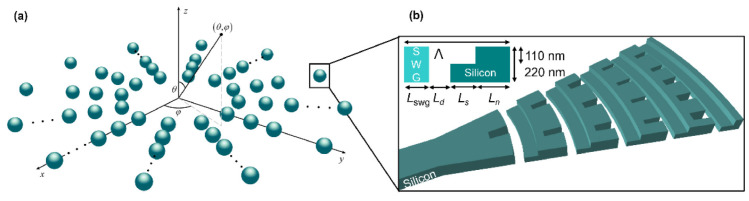
(**a**) Schematic of a two-dimensional circular optical phased array. The green spheres represent individual nano-antennas. (**b**) Three-dimensional schematic of a silicon-based surface-emitting antenna. Inset: Two-dimensional side view of the single antenna period with *L*-shaped radiating segments.

**Figure 2 sensors-22-06135-f002:**
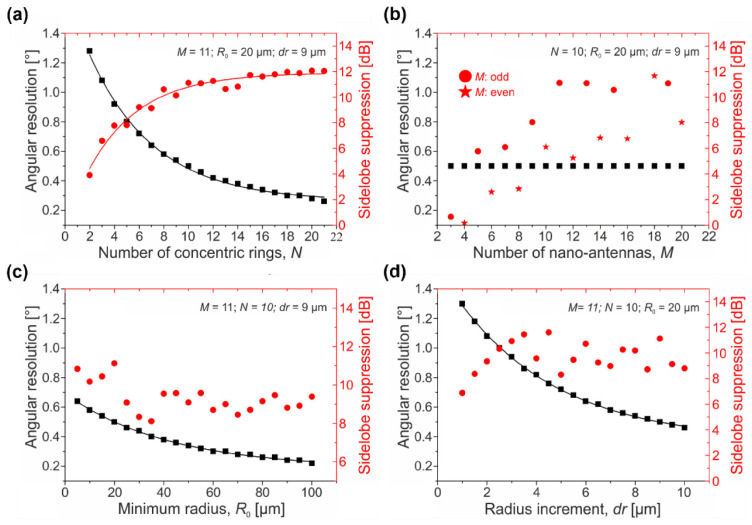
Angular beamwidth and sidelobe suppression of a circular array as a function of (**a**) number of concentric rings, *N*; (**b**) number of nano-antennas per ring, *M*; (**c**) minimum radius of rings, *R*_0_; and (**d**) delta radius between two adjacent rings, *dr*.

**Figure 3 sensors-22-06135-f003:**
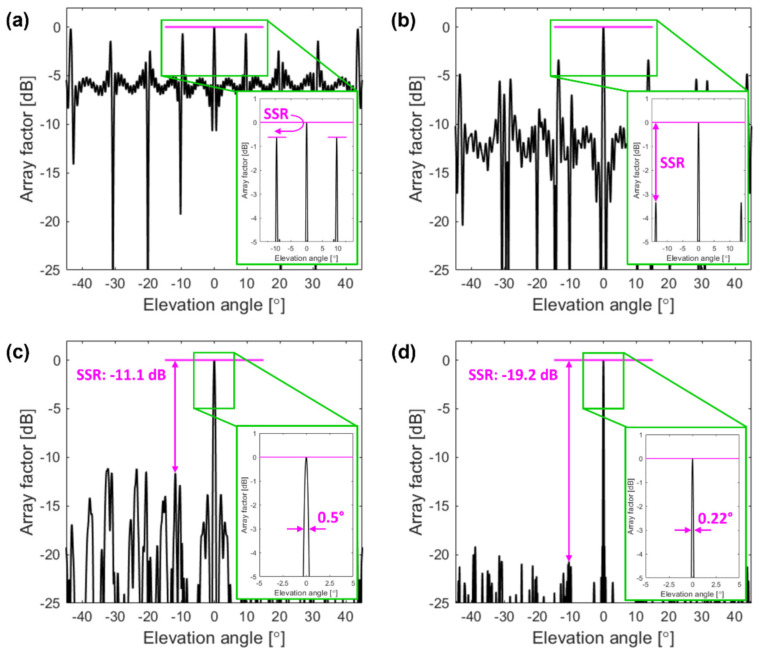
Simulated array factor for different optical phased arrays. Elevation cut (*Φ* = 0°) of the array factor amplitude for designs with (**a**) *N* = 10, *M* = 4, *R*_0_ = 20 μm, and *dr* = 9 μm; (**b**) *N* = 10, *M* = 8, *R*_0_ = 20 μm, and *dr* = 9 μm; (**c**) *N* = 10, *M* = 11, *R*_0_ = 20 μm, and *dr* = 9 μm (*Array #1*); and (**d**) *N* = 20, *M* = 41, *R*_0_ = 60 μm, and *dr* = 9 μm (*Array #2*). Figure insets show the enlarged view of an array factor with an elevation angle *θ* in a −5° to 5° range.

**Figure 4 sensors-22-06135-f004:**
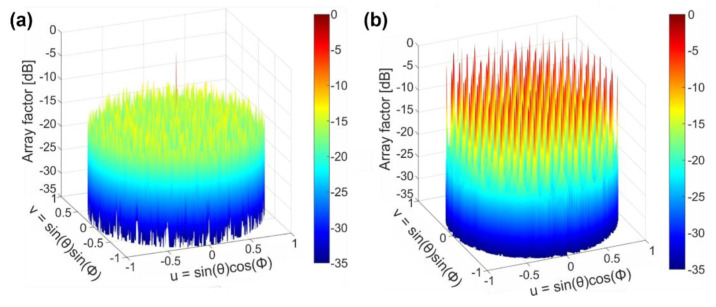
The array factor representation in a *uv* space. (**a**) Circular array: *N* = 10, *M* = 11, *R*_0_ = 20 μm, and *dr* = 9 μm at *θ* = 0°, *Φ* = 0°. (**b**) Rectangular array: *N* = 10, *M* = 11, *d_x_* = *d_y_* = 9 μm at *θ*_A_ = 0°, *Φ*_A_ = 0°.

**Figure 5 sensors-22-06135-f005:**
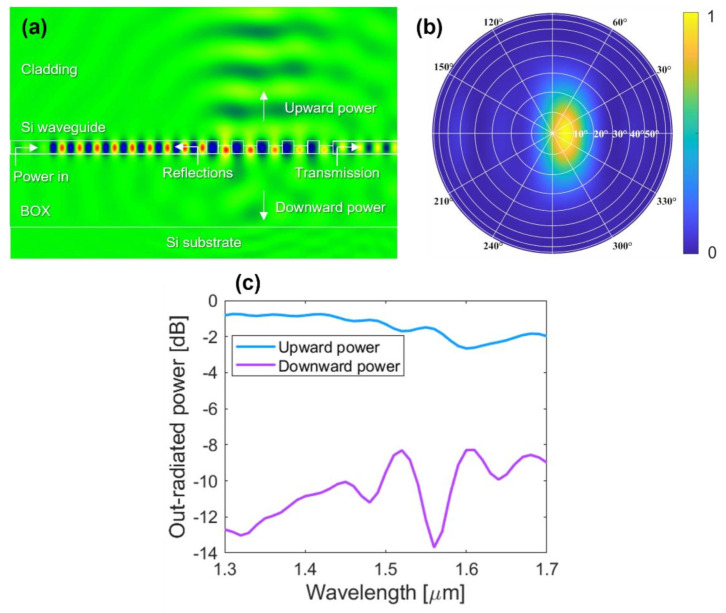
(**a**) Electric field distribution (the real part of *E*_y_ field component) of the *L*-shaped optical antenna. (**b**) The antenna far-field pattern, radiating at 1550 nm wavelength. (**c**) Out-radiated power (power up and down) from optical antenna as a function of the wavelength.

**Figure 6 sensors-22-06135-f006:**
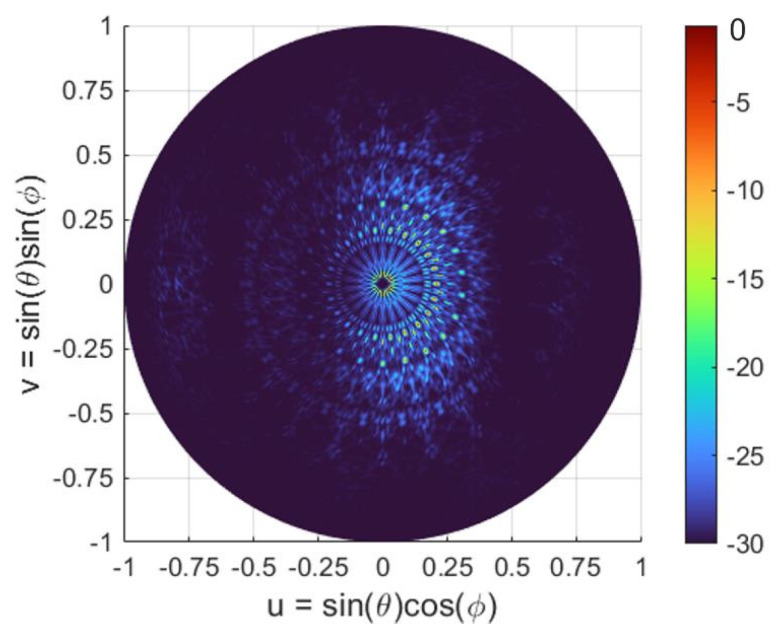
An array far-field pattern representation in a *uv* space for the *Array #1*, with the operating wavelength of 1.55 μm, and steering to *θ*_A_ = 0° and *Φ*_A_ = 0°. The peak beam is radiated at *θ* = *θ*_0_ = 6° and *Φ* = *Φ*_0_ = 0°.

**Figure 7 sensors-22-06135-f007:**
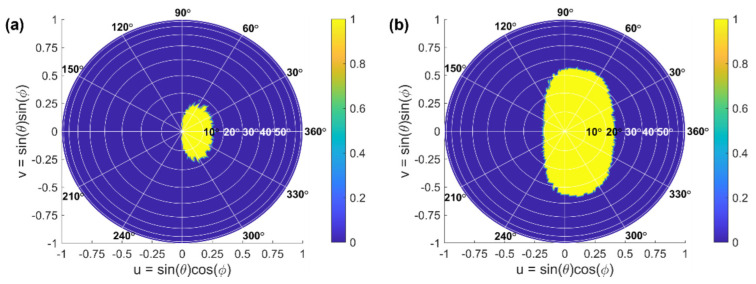
Projection of the steering range onto a hemisphere for (**a**) *Array #1* and (**b**) *Array #2*, both operating at 1550 nm wavelength. The bright area represents the set of all steerable angles, where the sidelobe suppression is larger than 10 dB, and the dark area represents non-steerable angles.

**Figure 8 sensors-22-06135-f008:**
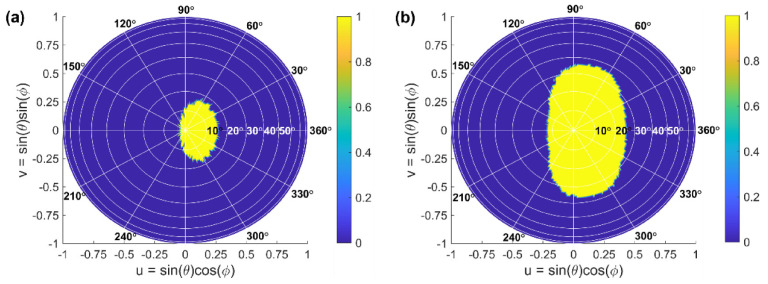
Projection of a steering range for (**a**) *Array #1* and (**b**) *Array #2*, both operating at *C*-band wavelengths. The yellow area represents the set of all steerable angles, where the sidelobe suppression is larger than 10 dB, and the blue area represents non-steerable angles.

**Figure 9 sensors-22-06135-f009:**
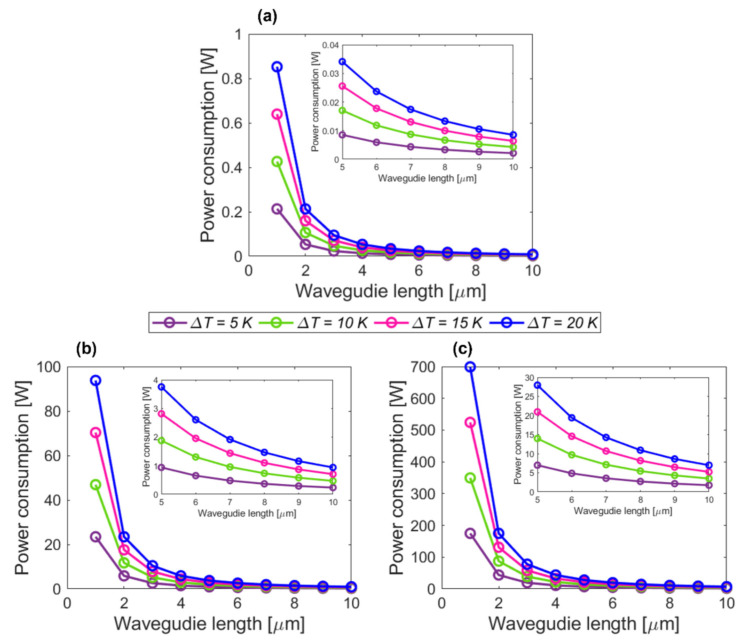
Power consumption analysis versus varied waveguide lengths to implement thermo-optic phase shifters, with a temperature changes Δ*T* in the 5 K to 20 K range. (**a**) Single thermo-optic phase shifter; circular OPA designs with (**b**) 110 and (**c**) 820 elements, respectively.

**Table 1 sensors-22-06135-t001:** Design parameters of optical phased arrays with circular geometry.

*Parameter/*	*N*	*M*	*R*_0_ [μm]	*dr* [μm]
*Design*				
** *Array #1* **	10	11	20	9
** *Array #2* **	20	41	60	9

**Table 2 sensors-22-06135-t002:** Passive photonic elements in the circular optical phased array.

*Component*	Central Coupler	Optical Antenna	Adiabatic Taper	Connecting Waveguide *
**Loss [dB]**	1.8	0.4	0.2	~0

* Note: Sub-dB waveguide loss per cm are assumed in this case, considering state-of-the-art. silicon-foundry-compliant processing [61]. In the proposed circular OPAs, short interconnecting strip waveguides are considered.

## Data Availability

Data are contained within the article.

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
