# Peer review of "Circular Optical Phased Array with Large Steering Range and High Resolution"

_sensors, 2022, doi:10.3390/s22166135_

Round 1
Reviewer 1 Report
The Authors showcase the design of a new 2D optical phased array based on a circular geometry, thus improving the signal-to-noise ratio and the resolution. The introduction is well written and gathers opportune references, the scientific procedure is sound and the results appear interesting also for people with are not in the field of sensors. Therefore, in my opinion, the present manuscript deserves publication in Sensors.
Reviewer 2 Report
The article required major revision before accepting.
In ref. [37] the authors have reported simulation result and numerical analysis for 2D optical phased arrays (OPAs) with radially arranged nano-antennas. The simulation result in that work shows suppression of the grating lobes, expanding the range for beam steering and obtaining narrower beamwidths, while increasing element spacing. In ref. [37] 110-element and 870element OPAs have been used.
In the proposed article the authors have reported the same 2D circular OPA with 110 and 820 elements. The authors need to explain how the recent work is significantly better than or differs from ref. [37]. Can the authors please explain this difference in the article?
In Fig. 1 how many phase shifters were used? Are the phase shifters heating based or PN junction based? What is the power consumption of the phase shifters? An explanation is also required how the beam steering was achieved using the phase shifters.
The proposed work is also a simulation and analysis work (similar to ref. 37), therefore a table should be added by referencing recent works on integrated OPAs from other research groups, and how the proposed solution is better in terms of loss, power consumption compared to other works.
Reviewer 3 Report
In this manuscript, the authors proposed two designs of circular optical phased arrays (OPA). The circular OPAs are promising since they can offer large steering angle while maintain high resolution simultaneously. The manuscript highlights the factors that determine the angular resolution and side lobe suppression by comparing two designs with different element numbers. The proposed designs are thoroughly investigated, and the simulation results are well explained. However, some design details seem missing. Although the authors may have published the details in their previous work, such as Ref [37], it would make this manuscript more self-contained if the details could be included rather than referred. For example, the feeding stage may intrigue readers, and yet this part is missing. In addition, the authors surveyed the literature extensively, and it might be helpful to incorporate the optical leaky wave antennas since they belong to the OPA category. Below are some example publications for optical leaky wave antennas.
Silicon-based optical leaky wave antenna with narrow beam radiation", Opt. Express, vol. 19, no. 9, pp. 8735-8749, Apr. 2011.
Experimental demonstration of directive Si 3 N 4 optical leaky wave antennas with semiconductor perturbations ", J. Lightw. Technol., vol. 34, no. 21, pp. 4864-4871, Nov. 2016.
An Optical Leaky Wave Antenna by a Waffled Structure", Journal of Lightwave Technology, vol.35, no.11, pp.2273-2279, 2017.
Graphene-based directive optical leaky wave antenna, Microwave and Optical Technology Letters, Vol. 61, Iss. 1, p. 153 (2019).
Round 2
Reviewer 2 Report
1. If the phase shifters are thermal, then a detailed analysis for thermal phase shifters on SOI platform is already provided in ref. [1]. You can use the data provided in ref [1] to estimate total power consumption of the device for the 110 and 820 elements. Also estimate the optical loss of the designed device. If comparison with other works on OPA is not possible this might be ok. But by reading the paper on your designed OPA, the readers should get an idea about power consumption and optical loss of the new design, which a very basic design criteria of any new devices.
[1] Maxime Jacques, Alireza Samani, Eslam El-Fiky, David Patel, Zhenping Xing, and David V. Plant, "Optimization of thermo-optic phase-shifter design and mitigation of thermal crosstalk on the SOI platform," Opt. Express 27, 10456-10471 (2019)
Reviewer 3 Report
The revision has addressed my comments. It is ready to be accepted.
Author Response
We thank the Reviewer for his/her prompt review of our revised manuscript as well as we thank for his positive final decision on our manuscript to be accepted.